# Processing of incomplete images by (graph) convolutional neural networks

**Tomasz Danel** [1]   **Marek Śmieja** [1]   **Łukasz Struski** [1]   **Przemysław Spurek** [1]   **Łukasz Maziarka** [1]

## Abstract

We investigate the problem of processing incomplete images by neural networks without replacing missing values. To deal with this problem, we first represent an image as a graph, in which missing pixels are entirely ignored. The graph image representation is processed using SGCN – a type of graph convolutional neural networks, which is a proper generalization of classical CNNs operating on images. On one hand, our approach avoids the problem of missing data imputation while, on the other hand, there is a natural correspondence between CNNs and SGCN. Experiments confirm that our approach performs better than analogical CNNs with the imputation of missing values on typical classification and reconstruction tasks.

## 1. Introduction

Learning from missing data is one of the basic challenges in machine learning and data analysis (Goodfellow et al., 2016). In a typical pipeline, missing data are first replaced by some values (imputation) and next the complete data are used for training a given machine learning model (McKnight et al., 2007). The above approach depends strictly on the imputation procedure – if we accurately predict missing values, then the other model that operates on competed inputs can obtain good performance. However, it is not obvious how to select imputation method for a given problem, because it is difficult to validate its performance in a real-life scenario. Thus, there appears a natural question: *can we learn from missing data directly without using any imputation at preprocessing stage?*

While it is difficult to answer this problem in general, a few approaches have already been designed for particular machine learning models (Dekel et al., 2010; Globerson &

[1]Faculty of Mathematics and Computer Science, Jagiellonian University, Łojasiewicza 6, 30-428 Kraków, Poland. Correspondence to: Tomasz Danel <tomasz.danel@ii.uj.edu.pl>, Marek Śmieja <marek.smieja@uj.edu.pl>.

*Presented at the first Workshop on the Art of Learning with Missing Values (Artemiss) hosted by the 37th International Conference on Machine Learning (ICML).* Copyright 2020 by the author(s).

Roweis, 2006). In (Chechik et al., 2008) a modified SVM classifier is trained by scaling the margin according to observed features only. In (Grangier & Melvin, 2010), the embedding mapping of feature-value pairs is constructed together with a classification objective function. Pelckmans et al. (2005) model the expected risk, which takes into account the uncertainty of the predicted outputs when missing values are involved. In a similar spirit, a random forest classifier is modified to adjust the voting weights of each tree by estimating the influence of missing data on the decision of the tree (Xia et al., 2017). The authors of (Hazan et al., 2015) design an algorithm for kernel classification that performs comparably to the classifier which has access to complete data. Goldberg et al. (2010) treat class labels as an additional column in the data matrix and fill missing entries by matrix completion. The work (Śmieja et al., 2018) shows how to generalize fully connected neural networks to the case of missing data given only an imprecise Gaussian estimate of missing data. Liu et al. (2018) introduce partial convolutions, where the convolution is masked and renormalized to be conditioned on only observed pixels.

In this paper we interpret the image as a graph, in which each node coincides with a visible pixel, while edges connect neighbor pixels. Since missing values are not mapped to graph nodes, we avoid the problem of missing data imputation. In order to efficiently process such an image representation, we use spatial graph convolutional neural networks (SGCN) (Danel et al., 2020). In contrast to typical graph convolutions (Kipf & Welling, 2016; Veličković et al., 2017), which consider graph as a structure invariant to rotations and translations, SGCN introduces a theoretically-justified mechanism to take into account spatial coordinates of nodes. More precisely, it has been proven that SGCN is a proper generalization of typical convolutional neural networks (CNNs) that operate on images, i.e. every convolutional layer can be represented as a spatial graph convolution. This fact allows us to think about SGCN as a type of CNNs, which is able, in particular, to process incomplete images without imputation.

To verify the introduced procedure, we consider MNIST (LeCun et al., 1998) and SVHN (Netzer et al., 2011) image datasets. Experimental results show that SGCN performs better than typical CNNs with imputations on the tasks of image classification and reconstruction in the setting of

missing at random (when the missingness pattern is conditionally independent of the unobserved features given the observations).

## 2. Graph-based model for processing incomplete images

**General idea.** Images can be interpreted as vectors (tensors) of fixed sizes. If the values of selected pixels are unknown, then the vector structure is destroyed. To recover this structure, we need to replace missing attributes with some values. Substituting unknown inputs carries the risk of introducing unreliable information and noise to initial data. This may have negative consequences on data interpretation as well as can decrease the performance of subsequent machine learning algorithms.

Our idea is to interpret incomplete image as a graph. Graphs represent a relational structure, in which the number of nodes and edges are not fixed. If some pixels in the image are unknown, then the corresponding graph contains less nodes, but the way it is processed does not change. In consequence, graph-based representation of incomplete images is more natural than using imputation.

It is well-known that CNNs are state-of-the-art feature extractors for images. However, as explained above, it is not obvious how to apply CNNs to incomplete data without replacing missing values. In this paper, we use SGCN, which is a type of graph convolutional networks (GCNs), that takes spatial coordinates of nodes into account. It is proved that SGCN can mimic any image convolution and, in consequence, SGCN is able to work comparably to CNNs using analogical network architecture (number of layers, size of filters, etc.).

**Graph-based representation of incomplete images.** Formally, the image is represented as a tensor $\boldsymbol{H} = (\mathbf{h}_{ijk}) \in \mathbb{R}^{n \times m \times l}$, where $n, m$ denote height and width of the image and $l$ is the number of channels. In the case of missing data, we do not have information about pixels values at some coordinates. Thus the incomplete image is denoted by a pair $(\boldsymbol{H}, J)$, where $J \subset \{1, \dots, n\} \times \{1, \dots, m\}$ indicates pixels which are unknown. In other words, $\mathbf{h}_{ijk}$ is unknown for every $(i, j) \in J$. Clearly, for a fully-observed image, $J = \emptyset$.

To construct a graph-based image representation, we create a node for every visible pixel of $\boldsymbol{H}$, i.e.

$$V = \{v_{ij} : (i, j) \in J'\},$$

where $J'$ is the set of indices of the observed components. The edge is defined only for nodes that represent adjacent pixels. Formally,

$$E = \{(v_{ij}, v_{pq}) : (i, j) - (p, q) \in \{-1, 0, 1\}^2\}.$$

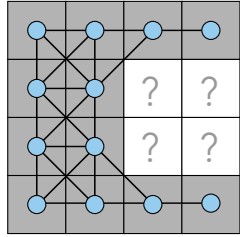

*Figure 1.* Graph construction for an incomplete image of the size $4 \times 4$ with a missing region of the size $2 \times 2$.

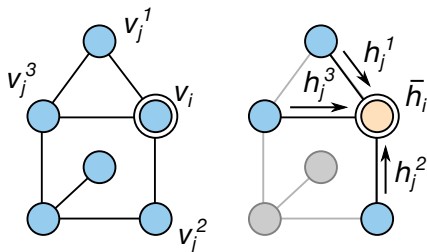

*Figure 2.* Basic idea of GCNs. Every filter is responsible for defining a pattern used to aggregate feature vectors from neighbor nodes.

Observe that for a complete image, every "non-boundary" pixel (node) has exactly 8 neighbors. In the case of incomplete data, the number of neighbors can be smaller, as the unknown pixels are not converted to nodes and, in consequence, the corresponding edge is not created, see Figure 1. The information about pixels brightness is supplied with a feature vector $\mathbf{h}_{ij} \in \mathbb{R}^l$ that corresponds to a node:

$$\boldsymbol{H} = \{\mathbf{h}_{ij} : (i, j) \in J'\}.$$

For a gray-scale image, $\mathbf{h}_{ij} \in \mathbb{R}$, while for a color picture $\mathbf{h}_{ij} \in \mathbb{R}^3$.

**Graph convolutions.** Let $G = (V, E, \boldsymbol{H})$ be a graph (representing the image $\boldsymbol{H}$) with $n$ nodes. To avoid multiple indexes in the following description, the node and the corresponding feature vector are denoted by $v_i$ and $\mathbf{h}_i$, respectively, while $e_{ij}$ is the edge between $v_i$ and $v_j$. To make a natural correspondence between graphs and images, we put $\mathbf{i} = \begin{pmatrix} i_x \\ i_y \end{pmatrix}$ to denote both pixel coordinates and index in graph.

Basic idea of GCNs is to aggregate the information of feature vectors from neighbor nodes over multiple layers, see Figure 2. To build a diverse set of patterns, GCNs use filters for defining a specific aggregation. Information from higher-level neighborhoods are fused by combining many layers together.

The above goal is realized by combining two operations. For each node $v_i$, feature vectors of its neighbors are first

aggregated:

$$\bar{\mathbf{h}}_i = \sum_{(v_i,v_j)\in E} U\mathbf{h}_j. \tag{1}$$

Observe that the aggregation is performed only over neighbor nodes, i.e $(v_i, v_j) \in E$. The weights $U \in \mathbb{R}^{O\times I}$ are either trainable (Veličković et al., 2017) or determined from a graph (Kipf & Welling, 2016), where $I$ and $O$ are the input and output sizes respectively. Next, a standard MLP is applied to transform the intermediate representation $\bar{H} = [\bar{\mathbf{h}}_1, \ldots, \bar{\mathbf{h}}_n]$ into the final output of a given layer:

$$\mathrm{MLP}(\bar{H}; W) = \mathrm{ReLU}(W^T\bar{H} + \mathbf{b}), \tag{2}$$

where a trainable weight matrix $W = [\mathbf{w}_1, \ldots, \mathbf{w}_n]$ is defined by column vectors $\mathbf{w}_i$. The dimension of $\mathbf{w}_i$ determines the dimension of the output feature vectors. A typical GCN is composed of a sequence of graph convolutional layers (described above). Finally, its output is aggregated to the network response depending on a given task, e.g. node or graph classification.

**Spatial graph convolutions.** In contrast to typical GCNs described above, SGCN uses spatial coordinates of nodes. In the case of images, spatial coordinates allows to identify a given pixel in the image grid, which is not possible using only the information about neighborhood. What is more important, the convolution defined by SGCN is constructed so that it is able to reflect any convolutional filter of typical CNNs. In other words, any image convolution can be obtained by a specific parametrization of SGCN. This makes a natural correspondence between SGCN and CNNs. This property cannot be obtained by simply adding spatial coordinates to feature vectors in classical GCNs.

From a formal side, SGCN replaces (1) by:

$$\bar{\mathbf{h}}_i(U, \mathbf{b}) = \sum_{(v_i,v_j)\in E} \mathrm{ReLU}\left(U\left[\begin{pmatrix}j_x\\j_y\end{pmatrix} - \begin{pmatrix}i_x\\i_y\end{pmatrix}\right] + \mathbf{b}\right) \odot \mathbf{h}_j, \tag{3}$$

where $U \in \mathbb{R}^{I\times 2}$, $\mathbf{b} \in \mathbb{R}^I$ are trainable, and $I$ is the dimension of the previous layer vectors. The pair $(U, \mathbf{b})$ plays a role of a convolutional filter which operates on the neighborhood of $v_i$. The operator $\odot$ is element-wise multiplication. The relative positions in the neighborhood are transformed using a linear operation combined with non-linear ReLU function. This scalar is used to weight the feature vectors $\mathbf{h}_j$ in a neighborhood. By the analogy with classical convolution, this transformation can be extended to multiple filters. Let $U = [U_1, \ldots, U_k]$ and $B = [\mathbf{b}_1, \ldots, \mathbf{b}_k]$ define $k$-filters. The intermediate representation $\bar{\mathbf{h}}_i$ is a vector defined by:

$$\bar{H}_i = \left[\bar{\mathbf{h}}_i(U_1, \mathbf{b}_1), \ldots, \bar{\mathbf{h}}_i(U_k, \mathbf{b}_k)\right].$$

Finally, MLP transformation is applied in the same manner as in (2) to transform these feature vectors.

## 3. Experiments

We evaluate our model on two machine learning tasks: classification and reconstruction of incomplete images.

For a comparison, we use **vanilla GCN**, which is one of the simplest GCNs that ignores spatial coordinates[1]. Moreover, we combine a typical **CNN** with various types of imputations: (i) **mask**, which is a zero imputation with an additional binary channel indicating unknown pixels (ii) **mean** imputation, where absent attributes are replaced by mean values for a given coordinate (iii) **k-nn** imputation, which substitutes missing features with mean values of those features computed from the k-nearest training samples (we use $k = 5$). For a fair comparison, every architecture (GCN and CNN) has the same structure, i.e. number of layers and filters. In the mask variant, the input to the first layer has 2 channels while other variants have only one – the following layers have the same number of input and output channels. The inner layers, with 64 input and 64 output channels, have: 38 656 parameters in SGCN, 36 928 in CNN, and 4 160 in GCN.

**Classification.** In this experiment, we use gray-scale handwritten digits retrieved from MNIST database and color house-number images of the SVHN dataset. For each MNIST image of the size $28\times 28$, we remove a square patch of the size $13 \times 13$. The location of the patch is uniformly sampled for each image. In the case of SVHN images of the size $32 \times 32$, we use patches of the size $15 \times 15$. This setting corresponds to the features missing at random.

Classification network is composed of 8 convolutional layers. Each one contains 64 filters of the size $3 \times 3$. Batch normalization is used after every convolutional layer. As mentioned, we use analogical architecture for both graph convolutions and typical image convolutions. We report test errors.

It is evident from Table 1 (first two rows) that SGCN performs significantly better than the other version of GCN. It is not surprising because, in contrast to typical GCNs, SGCN introduces information about spatial coordinates to the model. Next observation is that SGCN gives lower errors than CNNs combined with imputation strategies. While the advantage of SGCN over the second best method in the case of MNIST is slight, the difference is higher in the case of SVHN, which is significantly harder dataset to classify. As can be seen, the knowledge about missing pixels is crucial for the success of CNNs. Indeed, CNN (mask) gives higher accuracy than using imputation strategies[2]. In contrast to CNN (mask), which uses an additional binary

---

[1]We also experimented with graph attention network (Veličković et al., 2017), but the results did not improve.

[2]We additionally verified that combining masking with mean/k-nn imputation does not lead to further improvement of CNNs.

*Table 1.* Classification error on two incomplete images.

| Dataset | SGCN | GCN | CNN (mask) | CNN (mean) | CNN (k-NN) |
|---------|------|-----|------------|------------|------------|
| MNIST | **4.6%** | 31.4% | 4.9% | 5.9% | 5.7% |
| SVHN | **16.6%** | 74.6% | 18.6% | 19.9% | 22.4% |

*Table 2.* Classification error on complete images.

| Dataset | SGCN | GCN | CNN |
|---------|------|-----|-----|
| MNIST | **0.4%** | 16.3% | 0.5% |
| SVHN | **4.3%** | 75.4% | 5.1% |

channel to pass the information about unknown values to the neural model, SGCN directly ignores missing pixels, which is more natural. We also verified that SGCN gives satisfactory performance on complete images (no missing values), Table 2. While the difference between SGCN and CNN is slight, the test error of vanilla GCN is still very low. An important thing is that the disproportion between SGCN and CNN is higher for incomplete data than for complete images, which suggests that our strategy for dealing with missing values is beneficial.

**Reconstruction.** Reconstruction of incomplete images finds applications in image inpainting as well as is useful in restoring partially destroyed or low-quality images. In this experiment, we consider images taken from MNIST dataset and use the same size of removed patches as before.

We consider the auto-encoder architecture (AE). In the case of graphs, the encoder is implemented as SGCN with 5 spatial convolutional layers while the decoder is a simple deconvolutional neural network, which returns the image in the form of tensor. For imputation methods, we use a standard convolutional AE with identical number of layers and filters as in the case of our model. We assume that the complete data are not available in training phase. Therefore, for all models, the loss is defined as the mean-square error (MSE) calculated outside the missing region.

It can be seen from the Figure 3 that SGCN gives similar results to CNN (mask). The reconstructions coincide on average with ground-truth and are free of artifacts. There was a problem in restoring digit "9" (last row), but the same holds for other methods. The results produced by CNN (mean) and CNN (k-NN) are sometimes blurry. In contrast to CNNs, our method is more stable, because it does not depend on imputation strategy. In consequence, it may give worse results than CNN when it is easy to predict missing values, but, at the same time, it should perform better if the imputation problem is more difficult. Another advantage is that SGCN is trained end-to-end (no preprocessing of missing values).

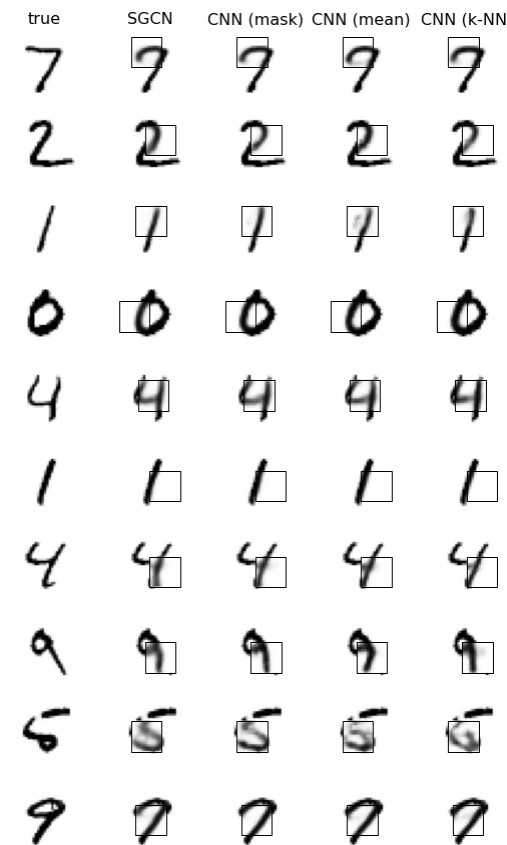

*Figure 3.* Reconstructions obtained for MNIST dataset (the first 10 images of test set).

## 4. Conclusion

We presented an alternative way of processing incomplete images by neural networks, which does not require replacing missing values at preprocessing stage. While graph representation of incomplete images avoids the problem of imputation, applying SGCN allows us to reflect the action of classical CNNs. The applied graph-based approach gives at least as good performance as typical CNNs combined with imputation strategies. The main disadvantage of this approach is the computational cost of using GCNs. In contrast to classical CNNs, the current implementations of GCNs are less efficient and it is difficult to process high dimensional images by very deep neural networks.

## Acknowledgements

This work was partially supported by the National Science Centre (Poland) grant no. 2019/33/B/ST6/00894, 2018/31/B/ST6/00993, and 2017/25/B/ST6/01271.

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
