# OpenReview forum: "Processing of incomplete images by (graph) convolutional neural networks"
_ICML.cc/2020/Workshop/Artemiss — ICML Artemiss 2020_

### Official Review · AnonReviewer1 · 2020-06-24
**New way of ingoring missing values for convnets**

**Rating:** 8
**Confidence:** 4

**Review:**

The authors propose a way to train convnet-like architectures with imputs with missing values.

The idea is to see the image as a possibly incomplete graph, and to leverage a recent advance in graph neural nets (called Geo-GCN) that allow to take spacial coordinates into account while applying graph convolutional filters.

The approach seems quite elegant and potentially very useful, and discussing about this paper would be interesting for the workshop.

A few comments:

- I don't fully grasp why regular convnets are particular cases of the proposed Geo-GCN. In particular, do you still get some of the nice properties of vanilla convolutions like equivariance?
- An experiment on complete data would also be interesting. It would be nice to check that the Geo-GCN does not have much worse resutls than a plain convnet
- You say that every architecture has the same structure, but it looks like they will have slightly different numbers of parameters (e.g. the "mask" approach has an additional input channel). Providing the exact numbers of parameters can be insightful.
- It would be interesting to talk about under what kind of missingness assumption the approach work. It's clear that it is mostly adapted to cases where the missing data are ignorable, e.g. missing (completely ?) at random. A discussion on this would be interesting.

---

### Decision · Program_Chairs · 2020-07-02

**Decision:**

Accept

**Comment:**

We're happy to accept this paper at Artemiss. We'll contact you soon to inform you about more details concerning the format of your presentation at the workshop, and the camera-ready version deadline. Please take into account the referee's comments to write the camera-ready version.